# Biofilm Formation in Medically Important *Candida* Species

**DOI:** 10.3390/jof9100955

**Published:** 2023-09-22

**Authors:** Zuzana Malinovská, Eva Čonková, Peter Váczi

**Affiliations:** Department of Pharmacology and Toxicology, University of Veterinary Medicine and Pharmacy, Komenského 73, 041 81 Košice, Slovakia; eva.conkova@uvlf.sk (E.Č.); peter.vaczi@uvlf.sk (P.V.)

**Keywords:** *Candida*, biofilm, cell, adherence, gene

## Abstract

Worldwide, the number of infections caused by biofilm-forming fungal pathogens is very high. In human medicine, there is an increasing proportion of immunocompromised patients with prolonged hospitalization, and patients with long-term inserted drains, cannulas, catheters, tubes, or other artificial devices, that exhibit a predisposition for colonization by biofilm-forming yeasts. A high percentage of mortality is due to candidemia caused by medically important *Candida* species. Species of major clinical significance include *C. albicans*, *C. glabrata*, *C. tropicalis*, *C. parapsilosis*, *C. krusei,* and *C. auris*. The association of these pathogenic species in the biofilm structure is a serious therapeutic problem. *Candida* cells growing in the form of a biofilm are able to resist persistent therapy thanks to a combination of their protective mechanisms and their ability to disseminate to other parts of the body, thus representing a threat from the perspective of a permanent source of infection. The elucidation of the key mechanisms of biofilm formation is essential to progress in the understanding and treatment of invasive *Candida* infections.

## 1. Introduction

Biofilm formation by various microbes is a critical problem that has been increasingly appearing in both human and veterinary medicine in recent years. A biofilm is defined as a structured microbial community interlocked by a protective extracellular matrix and attached to a biotic or abiotic surface. The biofilm can be formed by a single species or a mixed culture of bacteria and yeast [1]. Biofilms can initiate or prolong infections by providing a safe haven that resists treatment and from which cells can invade local tissue and simultaneously establish new sites of infection. They can also lead to the failure of implanted medical devices [2].

The formation of *Candida* biofilms has been observed on multiple surfaces, including living surfaces (mucous membranes, organs, blood vessels) and non-living surfaces, most often in the case of medical devices that come into contact with patients’ bodies (stents, shunts, implants, endotracheal tubes, pacemakers, and multiple types of catheters) [3,4,5]. Species of the genera *Candida* and *Malassezia* are among the medicinally important yeasts that have the ability to form biofilms [5,6]. Systemic mycoses are very closely related to the ability to form a biofilm. The latest statistics indicate approximately 1.6 million human deaths occur due to systemic mycoses each year. The opportunistic yeasts of the genus *Candida* are responsible for 90% of all these diseases and are also the fourth most common cause of all nosocomial blood infections [7,8,9]. The genus *Candida* consists of more than 150 species, of which only a few are considered to be pathogens causing disease in humans and animals. This is because up to 65% of *Candida* species are not able to grow at a temperature of 37 °C, limiting their ability to be successful pathogens or true commensals of humans. The species *C. albicans* is among the most important and most frequently isolated from healthy and diseased persons and is considered a commensal organism and, at the same time, a pathogen causing health complications. *C. albicans* represents over 80% of isolates from all forms of human candidosis [10]. Other pathogenic species include *C. glabrata* (*Nakaseomyces glabrata* in the new nomenclature), *C. tropicalis*, *C. parapsilosis*, *C. krusei* (*Pichia kudriavzevii* in the new nomenclature), and *C. auris,* which are referred to as non-*albicans* species. The number of infections caused by non-*albicans* species is increasing [11,12,13,14,15]. The relatively new species, *C. auris*, also belongs to the group of dangerous nosocomial pathogens [13,16]. Due to the often inaccurate biochemical identification of *C. auris* by commercial laboratories, it is necessary to use molecular methods for confirmation. After its first identification, only a few cases of *C. auris* were confirmed from misidentified samples from the past, namely in Pakistan and the Republic of Korea [17]. Thousands of stored *Candida* isolates from the SENTRY isolate collection were examined with negative results, which confirmed that this is a newly emerged species and that it has spread around the world only in the last 14 years [16,18].

Although the *Candida* genus can exist as the normal flora of mucous and skin membranes, under certain conditions, it becomes pathogenic. *Candida* yeasts show a stronger binding affinity to mucous membranes than to skin. They are the main etiological agents of infections of the oral cavity, gastrointestinal, respiratory, and urogenital tract; the most common are oral candidosis and vulvovaginal candidosis with the presence of a biofilm [8,19,20]. They can also cause serious disseminated bloodstream infection (candidemia) and the infection of tissues or deep organs (candidiasis) [21]. *Candida* yeasts represent a risk especially in the hospital environment, i.e., in the intensive care unit, where they appear as nosocomial pathogens with high morbidity and mortality in the chronically ill and in post-operative patients. Due to their ability to adhere to non-living surfaces of medical devices introduced into the patient’s body (stents, shunts, implants, endotracheal tubes, multiple types of catheters, pacemakers, artificial heart valves, joint replacements, and more) and colonize them, *Candida* yeasts represent a notable medical problem. The ability to form a biofilm appears as one of the *Candida* virulence factors in the development of fungal infection [3,4,5,15,22,23]. Unlike other pathogenic species, which often occur as natural commensals of the human body, *C. auris* is transmitted very often from person to person or from contaminated environments, equipment, and tools [24,25,26]. Laboratory studies have confirmed its survival on contaminated surfaces in the environment from 7 days to 4 weeks [27,28]. In a study monitoring the survival of *C. auris* planktonic cells or cells in the form of a biofilm on glass, plastics, fabrics, steel, and wood, despite daily disinfection, its survival was found to be 3 weeks [29].

The following approximate occurrence of *Candida* species was reported in Europe and the United States: *C. albicans* (50%), *C. glabrata* (30%), *C. parapsilosis* (12%), *C. tropicalis* (7%), and *C. krusei* (1%) [30]. According to a study tracking the occurrence of non-*albicans* species over 20 years, the most frequently identified species are *C. glabrata* (18.7%), *C. parapsilosis* (15.9%), *C. tropicalis* (9.3%), and *C. krusei* (2.8%) [31]. In India, *C. tropicalis* (41.6%) was reported with the highest presence in 1400 patients treated in the ICU. Among the other species, *C. albicans* (20.9%), *C. parapsilosis* (10.9%), *C. glabrata* (7.1%), and *C. auris* (5.2%) were detected [32]. The prognosis of *Candida* infections depends on the species diagnosed, the type of isolate, whether it is a planktonic or biofilm-forming *Candida*, the subsequent determination of susceptibility or resistance, and the health status of the patient. It is necessary to focus on differentiating biofilm-forming isolates because the mortality rate caused by biofilm formation in *Candida*-related infections has been shown to be increased almost two-fold when compared to planktonic infections [33]. A meta-analysis of 31 studies reported a pooled attributable mortality of 37.9% with planktonic cells while the mortality associated with biofilm-forming strains was 70.0% in *Candida*-related bloodstream infections [34]. This study reported that *C. tropicalis* was the most prevalent species among the biofilm-forming organisms (67.5%) in candidemia, even more so than *C. albicans* (30.3%). In *C. krusei*, lower percentages of causation but high mortality were reported. Some studies have confirmed high mortality in *C. krusei* (40–59%), especially in patients with neutropenia and hematologic malignancies [35,36]. *C. krusei* belongs to the group of very important pathogenic species, but it is understood the least [37]. Since the first identification of *C. auris* in 2009, it has been confirmed in a variety of infections involving the ear, internal organs, brain, and bones and causes invasive infections associated with multidrug resistance and high mortality, from 30% to 72% [13,16]. *C. auris* was found to be able to evade the immune response (neutrophils) and stimulate the innate immune response of the infected organism [38,39]. The World Health Organization decided to include it in the critical fungi group [29]. Information regarding the percentage representation of individual species within candidiasis and candidemia varies greatly; however, it is important to note that all medically important *Candida* species pose a serious risk.

## 2. Biofilm Formation

Biofilm formation is influenced by several factors and mechanisms on the part of the yeast and the host organism. *Candida* yeast can form a biofilm in 38–72 h. It is a complex process that consists of several stages, the first of which is the adhesion of the yeast to a living or non-living surface. The next stage of biofilm development consists of cell proliferation and the early stage of adhered cell filamentation. This is followed by the maturation of the biofilm. The final step of biofilm development is the dispersion stage, in which some yeast cells disperse from the biofilm into the environment. The process of biofilm formation varies in part depending on the *Candida* species. The ability of *Candida* yeasts to proliferate and to establish biofilms is also influenced by their interaction with host homeostasis and variation (mucosal pH shifts or nutritional changes) and the state of the host’s immune system [2,40].

### 2.1. Adhesion

During the adhesion phase, solitary cells adhere to a suitable surface, aggregate into microcolonies, and form a basal monolayer. The adhesion process takes approximately 11 h (in vitro) [40,41]. Each *Candida* species has its own specifics. *C. albicans* is able to adhere much more effectively to the epithelial cells of the gastrointestinal tract, urogenital tract, and endothelium of blood vessels than, e.g., *C. glabrata*. *C. glabrata* has a better ability to adhere to platelets, which allows it to distribute well in the bloodstream, especially in disseminated infection [42]. *C. tropicalis* is more virulent in neutropenic hosts, usually with hematogenous dissemination to peripheral organs. *C. tropicalis* is also a frequent colonizer of the upper respiratory tract and forms biofilms on tracheal tubes [43,44]. *C. parapsilosis* is able to form tenacious biofilms on central venous catheters and often threatens undernourished children and low-birth-weight neonates [45,46].

Adherence is mediated by adhesins—glycoproteins located on the surface of the cell wall. The adhesins are encoded by different genes whose expression occurs in different phases of yeast cell growth and development. Fungal adhesins provide for interactions between cells (flocculation, filamentation), between a cell and an inert surface (e.g., agar, plastic material), and with host tissues [42,47,48]. In *Candida*, the adhesion ability is encoded by several gene families. Large differences in genes and their regulation are noted; in addition, the gene variability depends on the yeast species. There are several levels of variability, including strain-specific and allele-specific differences in size for the same gene, strain-specific differences in gene regulation, and the absence of specific genes or gene regions in certain isolates. Most typical for *C. albicans* is the *ALS* gene family, encoding large cell-surface glycoproteins that are implicated in the process of adhesion to host surfaces. *ALS* genes also occur in other *Candida* species, e.g., *C. parapsilosis*, *C. tropicalis*, *C. krusei,* and *C. auris* [37,49,50,51]. In *C. glabrata*, a main group of adhesins is encoded by the *EPA* (epithelial adhesin) gene family [52].

Other factors can also affect the adhesive property. One of these is cell surface hydrophobicity, which increases the adhesion of cells to surfaces, but it varies from species to species. The greatest association between hydrophobicity and better adhesion was demonstrated in *C. parapsilosis* and *C. tropicalis* [53]. In *C. krusei*, a higher hydrophobicity of the cell wall was detected in comparison to *C. albicans*, and a correlation between hydrophobicity and adhesion to HeLa cells was confirmed [37]. The fibrillar layer of the cell wall is the site of primary contact between the pathogen and the host surface. It consists of glycoproteins, the majority of which belong to mannoproteins. Glycosylated mannoproteins of the outer layer of the cell wall, especially the acid-labile fractions of phosphomannoproteins, are responsible for the wall’s hydrophobic character [37,54]. Some *C. auris* strains have a specific property related to cell aggregation; however, some strains do not have this feature [55,56]. There is little information on how aggregation ability affects the pathogenicity of *C. auris*. A study in mouse models showed that aggregates can accumulate in organs after infection [57], but non-aggregative *C. auris* strains are more pathogenic, which may be related to the easier spreading of individual cells [58].

The substrate is one of the important factors that influences *Candida* yeasts’ adhesion and the subsequent formation and size of the biofilm. Several artificial materials suitable for biofilm formation and monitoring have been tested. The materials preferred by *C. albicans* were latex, silicon, elastomer, and teflon. *C. krusei* was capable of forming biofilms on polyethylene, polyvinylchloride, glass, and polystyrene surfaces [37]. Additionally, other *Candida* species may form biofilms on different artificial substrates, preferring teflon (*C. parapsilosis*), teflon and polyurethane (*C. tropicalis*), or polyvinylchloride (*C. glabrata*) [59]. The ability of *C. auris* to persist on contaminated surfaces in the form of a biofilm or cells and the possibility of transmission from latex and nitrile gloves to sterile urinary catheters points to a very serious problem in hospitals [29,60].

### 2.2. Proliferation and Maturation of Biofilm

Morphological modifications happen during the proliferation of biofilms (the intermediate phase). The number of cells increases and cell filamentation occurs. The formation of a biofilm depends on the ability of the yeast to produce extracellular polymeric substances (EPSs) such as polysaccharides, glucose, hexosamine, lipids, proteins, phosphoric acid, uronic acid, and others. The proliferative process can take approximately 12–30 h. Microcolonies form on the newly formed monolayer, followed by the differentiation of macrocolonies containing yeast cells, germ tubes, and young hyphae. The maturation phase is completed approximately within 72 h from the start of biofilm formation. Maturation results in a complex network of several layers of polymorphic cells, including hyphal cells (chains of cylindrical cells), pseudohyphal cells (ellipsoidal cells joined end-to-end), and round yeast cells encased in an extracellular matrix. During the organization of dividing cells, numerous pores and water channels are formed, through which the smooth circulation of molecules between the cells in the biofilm and the surrounding environment is ensured [34]. The resulting 3D structure is formed by a dense network of yeast and filamentous cells embedded in an extracellular matrix (ECM) formed by an exopolymeric material. EPS production depends on the carbon source and ensures the integrity of the biofilm and, among other functions, protects cells from phagocytosis and drug diffusion. It was detected that EPSs consist of up to 40% polysaccharides and their production also depends on the medium flow rate [61].

After the formation of a complete biofilm, a dispersal phase follows, during which some round daughter cells disperse from the biofilm into the environment, where they colonize other surfaces. The biofilm poses a serious threat in terms of a permanent source of infection [62].

Some differences in *Candida* biofilm formation were found when comparing in vivo and in vitro models. In a study investigating *C. albicans* biofilm properties, it was found that in vivo conditions were more favourable for faster biofilm formation. The maturation phase of the *C. albicans* biofilm in vivo was confirmed at 24 h in the rat central venous catheter and in an avascular implantation of a small catheter in rats, in contrast to the 38–72 h observed in in vitro models [63,64]. With *C. glabrata* it was the opposite, biofilm formation in vitro took less time than under in vivo conditions [65].

## 3. Characterization of *Candida* Species Biofilms

Within *Candida* species, there are large differences in the structure and properties of biofilms. The formation of a biofilm depends on many factors, not only on the type of yeast itself, but also on the surface suitable for adhesion and on the environment and its composition, including the pH, oxygen content, and flow rate [61,66,67]. Concentrations of metal ions in the surrounding environment influence growth and biofilm formation in *Candida* species. Metal cations Co^2+^, Cu^2+^, Ag^+^, Cd^2+^, Hg^2+^, and Pb^2+^ and anions AsO^2−^ and SeO_3_^2−^ inhibit hyphal formation in biofilms of *C. albicans* and *C. tropicalis*, whereas CrO_4_^2−^ triggers a transition to the hyphal cell morphotype in *C. tropicalis* biofilms [67]. It was shown that the increased flow rate of the medium significantly positively affects the formation of the matrix [61].

Of all medically important *Candida* species, *C. albicans* is considered the largest and most important biofilm producer. *C. albicans* is polymorphic with the ability to create true hyphae and, more frequently, pseudohyphae (Figure 1). True hyphae grow from yeast cells or arise as branches of existing hyphae. Pseudohyphae are formed from yeast cells or hyphae by budding. The daughter growth (pseudohyphae) remains attached to the parent cell and elongates, resulting in filaments with constrictions at cell–cell junctions, but without the internal dividing wall (septum) typical of true hyphae [68].

Biofilms formed by *C. albicans* have a more heterogeneous structure, consisting of cells, hyphae, and pseudohyphae surrounded by an ECM material (Figure 2). The ECM is important in terms of the network connection between cells with different surfaces and the connection between the cells themselves, and, at the same time, it creates a barrier between the biofilm and the surrounding environment. The ECM in *C. albicans* biofilm consists of 55% proteins and their glycosylated counterparts, 25% carbohydrates (mostly glucose and hexosamine), 15% lipids (mostly neutral glycerolipids and polar glycerolipids, and, to a lesser extent, sphingolipids), and 5% noncoding DNA [61,69]. Polysaccharides, including α-1,2-branched α-1,6-mannans with unbranched α-1,6-glucans and mannan, are associated with an important complex of the ECM [69,70]. In the biofilm structure, there are microcolonies surrounded by water channels [68]. *C. albicans* generally produce larger and more complex biofilms than other species [71].

*C. glabrata* is not polymorphic, does not form true hyphae and pseudohyphae, and its biofilm is formed exclusively by yeasts in the form of cells bounded by a multi-layered structure or in clusters forming a dense network [54]. The ECM of *C. glabrata* biofilm is composed of a high content of carbohydrates and proteins [72]. In a study monitoring *C. glabrata* biofilm formation in serum-coated triple-lumen catheters in a rat subcutaneous model, biofilm maturation occurred after 48 h, while the amount of biofilm reached a maximum after 6 days. The measured thickness of the *C. glabrata* biofilm was 75–90 ± 5 µm, which is approximately half of the *C. albicans* biofilm thickness. In general, *C. glabrata* cells (1–4 µm) are also smaller than *C. albicans* cells (4–6 × 6–10 µm) [65,72].

*C. parapsilosis* is not able to produce true hyphae but can form pseudohyphae, which are characterized by a large and curved appearance. *C. parapsilosis* biofilm forms when clusters of yeast cells (2.5–4 × 2.5–9 µm) and pseudohyphae adhere to a surface and combine into compact or discontinuous multilayer aggregates. Compared to other species, the amount of ECM in *C. parapsilosis* biofilm is low and contains mainly carbohydrates with low protein content [72,73,74].

*C. tropicalis* forms a biofilm in the model of a dense network of yeasts and pseudohyphae (hyphae in some strains) surrounded by ECM with a low content of carbohydrates and proteins. In comparison to other medicinally important *Candida* species, *C. tropicalis* cells are relatively large (4–8 × 5–11 µm) [61,66].

Unlike the other species, which have spheric or ovoid cell shapes, *C. krusei* is typically characterized by cylindrical yeast cells that may be up to 25 µm in length. *C. krusei* is able to create hyphae and pseudohyphae and, like *C. albicans*, shows thermodimorphism. *C. krusei* cells are capable of exclusively using glucose as a carbon source. Following adherence, *C. krusei* cells proliferate as yeast-form cells, and, subsequently, pseudohyphal and hyphal cells begin to form from these dividing yeast-form cells. These continue to elongate and proliferate throughout the completion of biofilm formation [37]. When monitoring isolates from Turkish patients with diagnosed candidiasis, it was found that approximately one-half were able to form a biofilm [75]. The presence of *C. krusei* antagonistically affects the development of *C. albicans* biofilm in an inhibitory manner related to the reduction in the expression of genes associated with adherence and transcriptional regulation of the biofilm formation process. A count reduction of 52.6% and 64.4% was observed for *C. albicans* at the 12 and 24 h time points [76]. The inhibitory effect of *C. krusei* on *C. albicans* was also confirmed by the findings in another study [77].

*C. auris* does not form hyphae and pseudohyphae, but under certain special cultivation conditions, it is able to grow into a pseudohyphae-like form [78,79]. *C. auris* cells grow singly, in pairs or groups, and have an ovoid, ellipsoidal to elongate shape that reaches 2.5–5.0 μm in size [80]. The extracellular matrix contains a large number of mannan-glucan polysaccharides which ensures drug sequestration and reduces the antifungal sensitivity of the cells; the sequestration can reach up to 70% of triazole antimycotics [70,81,82]. *C. auris* has been divided into five geographically and phylogenetically distinct clades (I, II, III, IV, and V) with different sensitivities to antifungal agents, which influences the survival of their biofilms in treated patients [18,83,84].

To a large extent, the formation of a biofilm, its structure, and its properties also depend on the strain itself and not only on the *Candida* species, which is a serious problem in terms of virulence and subsequent treatment [85].

## 4. Gene Regulation of *Candida* Biofilm Formation

Gene regulation of biofilms is very complex and still not fully understood. A large number of genes ensuring adhesion, hyphal formation, EPS production, proliferation, maturation, and dispersion regulate biofilm formation (Table 1). The presence of specific genes depends on the *Candida* species and on the specific strain, as well as on the environment, which largely influences the expression of specific genes required for the current phase of biofilm formation. Within the framework of biofilm gene regulation [86], *C. albicans* is the best studied [2,87]. The scientific field related to the expression of genes affecting biofilm formation in *C. krusei* and *C. auris* is unexplored.

The adhesion process of *Candida* yeasts involves more genes than those encoding adhesins. A group of ten genes (including *ALS1*, *ALS2*, *ALS4*, and *PGA6*) is upregulated early for the synthesis of adhesin proteins in biofilm formation, whereas a different set of ten genes (including *IFF4*, *IFF6*, *PGA32*, and *PGA55*) is upregulated at later time points [88]. More than 30 transcriptional gene regulators are required for *C. albicans* cell adhesion on silicone surfaces in vitro. It was confirmed that at least four of these regulators (*BCR1*, *ACE2*, *SNF5*, and *ARG81*) are also required for adherence to polystyrene surfaces [2]. No information is available on whether ALS proteins in *C. auris* have similar roles as in *C. albicans*. Rather, the research has focused on the ability of cells to aggregate; by comparing the upregulation of genes, differences were found between non-aggregative and aggregative *C. auris* cells during biofilm formation. A total of 450 genes were found to be upregulated in biofilm cells of the aggregative phenotype compared to the non-aggregative phenotype [58].

Some genes influence others. For example, *BCR1* is not necessary for hyphal morphogenesis; it acts as a positive regulator of the expression of cell surface proteins encoded by the *HYR1*, *ECE1*, *RBT5*, *ECM331*, *HWP1*, *ALS3*, *ALS1*, and *ALS9* genes, which are important for the adhesion process in *C. albicans* [89,90]. Some genes have a role in more than one phase of biofilm formation. The *HWP1* gene encodes a major *C. albicans* protein involved in several functions, including cell wall assembly, intracellular signalling, hyphal development, and binding of *Candida* cells to epithelial cells. In *C. albicans* interaction with *C. krusei* and *C. glabrata*, the *HWP1* gene was downregulated and the capacity of *C. albicans* to form biofilms was reduced [91]. In *C. albicans*, *HWP1* is influenced by the *BCR1* gene, which is regulated with the expression of *TEC1,* dependent on *EFG1* and *CPH2*. Transcription factor *TEC1* is an important regulator promoting filamentation in *C. tropicalis*, and a homologue of *TEC1* has also been found in *C. parapsilosis*. *EFG1* and *CPH1* positively control the expression of genes required for hyphal growth (*ECE1*, *HYR1*, *HWP1*, and *ALS3*) and the expression of cell wall-related genes due to the usual morphological alterations involved in the transition from yeast to hyphae. This was confirmed by the finding that the Δefg1 and Δefg1Δcph1 deletion mutants were incapable of filamentation or biofilm development, as they were only capable of ensuring a sparse monolayer of adherent elongated cells [59].

*BCR1* is a positive regulator of the *RBT1* gene and is responsible for the synthesis of the Rbt1 cell wall protein. This *RBT1* gene also positively influences a hyphally regulated cell wall protein and proteins from an agglutinin-like sequence protein family which are the most common proteins on *C. tropicalis* pseudohyphae, with confirmation also in *C. albicans* and in *C. parapsilosis* [92]. Efficient hyphae formation in *C. albicans* biofilms is affected by Zap1 (a zinc-responsive activator protein), which regulates the accumulation of farnesol [93]. Farnesol actively regulates the formation of hyphae in *C. albicans*, *C. tropicalis*, and *C. dubliniensis*. In addition, it discontinues biofilm creation in *C. parapsilosis*; however, hyphal growth is not affected by farnesol in this species. The increased farnesol concentration in the mentioned *Candida* species affects the regulation of the expression of certain genes that influence the synthesis of proteins ascertaining the structure of biofilms [94]. The *ZAP1* gene controls the expression of zinc transporters and other zinc-regulated genes, which are necessary for the formation of biofilms [93].

In the dispersal phase, yeast-form cells are released to colonize the surrounding environment. The gene regulation of this phase is the least explored. *PES1* is the only known molecular regulator that directly controls the production of lateral yeast cells from hyphae and the inducement of biofilm dispersal. *PES1* does not regulate hyphal morphology or biofilm architecture but is essential for sustained candidiasis. The inducer of dispersal *PES1* genetically interacts with the repressor of filamentation *NRG1* [95,96,97].

### Genes of Planktonic and Biofilm Cells

It was confirmed that up to 1524 genes are significantly changed when comparing gene expression between planktonic and biofilm cells. During dispersal cell formation, cells are primed for an invasive role before being released from the biofilm. Dispersed cells are developmentally different compared to biofilm and planktonic cells of the same age and are ready for immediate infection in the host organism. When analysing the genome of planktonic and dispersed cells, significant differences were noted in 963 genes. Many genes were found to be upregulated in dispersed cells [98]. The upregulation refers to genes of the *SAP* family, which in *Candida* ensure adhesion to host tissue, invasion, and damage or destruction of cells and molecules of the host’s immune system. In dispersed cells, a large number of genes were upregulated (*SAP3*, *SAP6*, *SAP8*, and *SAP9*), but in biofilm cells, some genes (*SAP4* and *SAP10*) also showed increased expression. The upregulation of lipase family genes was noted in *Candida* dispersed cells (*LIP3* and *LIP9*) and biofilm cells (*LIP4* and *LIP6*), while in planktonic cells, the expression of the *SAP* or lipase family genes was not confirmed [98,99]. Dispersed yeast cells express genes that are typical for the hyphal form. Although dispersed yeast cells have a yeast morphology and yeast-specific genes (e.g., *YWP1* and *RHD3*), they also express genes typical for the hyphal form (e.g., *DDR48*, *PHR2*, *ASC1*, *SUN41*, *ACE2*). It was confirmed that up to 167 upregulated and 168 downregulated genes are differentially expressed in dispersed cells compared to both biofilm and planktonic conditions [98].

## 5. Antimycotic Sensitivity of *Candida* Biofilm

Biofilm formation is an important virulence factor in a number of medically important *Candida* species, as it may lead to significant resistance to antifungal therapy. The biofilm limits the penetration of substances through the extracellular matrix and thereby prevents the effect of antifungals on all cells during treatment. The biofilm protects cells from the host’s immune responses and allows the hematogenous spread of cells, thus contributing to the spread of infection. The increasing resistance of *Candida* biofilm can be explained by several factors. One reason for the emergence of antifungal resistance of biofilms is that the EPSs act as a barrier to drug diffusion. Yeast cells in biofilms containing EPSs are 20% more resistant to amphotericin B compared to the same cells after the removal of the EPSs. In *C. auris*, the influence on drug diffusion is significantly greater. An important role in yeast resistance is played by β-1,3 glucan, whose increased amounts can be found in the biofilm matrix. The extracellular β-1,3-glucan matrix absorbs amphotericin B, thereby increasing the antifungal resistance of the cell. The absence of β-1,3 glucan in the matrix increases the sensitivity of *C. albicans* to fluconazole and amphotericin B [41,70,81,100].

Combinations of different resistance mechanisms often occur in clinical isolates of the genus *Candida*. Antimycotic resistance can be developed because of changes in the cell wall or plasma membrane leading to impaired drug absorption. Another cause is changes in the affinity to the drug target (e.g., 14-α-sterol demethylase in azole treatment) or in the cellular 14-α-sterol demethylase content due to target site mutation or overexpression of the *ERG11* gene. Azole antifungals are largely predominant in the therapy of systemic mycoses, whereas *C. krusei* is naturally resistant to fluconazole [101]. It was confirmed that C. tropicalis’ resistance to azole antifungal drugs increases [102]. Fluconazole is almost ineffective against *C. auris*; more than 90% of *C. auris* isolates are resistant to the drug. The most serious problem in the treatment of infections caused by *C. auris* is often multi-resistance to different groups of antifungals [103]. The mechanism of azoles’ action is based on the specific inhibition of 14-α-sterol demethylase, which catalyses the conversion of lanosterol to ergosterol. The enzyme is dependent on the action of fungal cytochrome P450. Azoles affect the metabolism of sterols in yeast cells and disrupt the synthesis of membrane lipids, thus having a fungistatic effect. In *Candida*, ergosterol (5,7-diene-oxysterol) is the most important structural lipid of cell membranes, ensuring their permeability and fluidity. Ergosterol is synthesized in the endoplasmic reticulum through the sequential activity of 25 different genes [104]. Currently, available antifungal drugs interfering with ergosterol synthesis affect the products of the *ERG11* gene (azoles), the *ERG1* gene (allylamines), and the *ERG2* gene (morpholines). There is great potential for other genes as antifungal targets. Potentially, all steps of ergosterol biosynthesis can be targeted by antifungal drugs, and this opens up new possibilities in the development of new antifungals [105].

Efflux pumps play a significant role in the mechanism of antimycotic resistance in *Candida* species. There are two types of efflux pumps located in the cytoplasmic membrane which transport antifungal drugs or other toxic substances (xenobiotics) from the cell. The first group includes primary transporters using energy from ATP degradation, which are proteins belonging to the ATP-binding cassette (ABC) family [106]. The second group consists of secondary transporters drawing energy from the concentration gradient of protons typical for biological membranes. This category is represented by MFS (major facilitator superfamily) proteins [107]. In the case of yeast, the *CDR1* and *CDR2* gene products are ABC proteins, and the *MDR1* products are MFS-type transporters. Changes in gene expression during *Candida* biofilm formation include upregulation of *CDR* and *MDR* genes encoding azole resistance. This regulation appears to be important for the development of antifungal resistance in the process of biofilm formation and persistence [106,107]. In *C. auris*, ABC and MFS efflux pumps were also confirmed to be involved in fungal resistance, and even *C. auris* has more activity in ABC-type efflux pumps than other species [108,109]. *C. auris* strains show a higher resistance to amphotericin B (mutations in *ERG* genes) after fluconazole (mutations in the *ERG11* gene), followed by reduced susceptibility to 5-fluorocytosine (mutations in the *FCY2*, *FCY1*, and *FUR1* genes) and caspofungin (mutation in the *FKS1* gene) [103].

The increase in antifungal resistance of clinical *Candida* isolates is a reason for caution in the overuse of antifungals, as there are few available treatments for severe *Candida* infections. Each treatment should be preceded by a determination of the sensitivity of the isolated strain of *Candida*. Pharmacological strategies in the therapy of unresponsive fungal infections include the use of new forms of antifungals, such as β-cyclodextrin itraconazole, amphotericin B lipid complex, and amphotericin B colloidal dispersion or combinations of several antifungals, e.g., fluconazole and flucytosine, amphotericin B and fluconazole, or caspofungin and fluconazole [110]. Potential alternative therapy involves the use of new active substances obtained from various general sources, such as natural products, synthetic substances, or polymeric materials, which have been shown to be active in vitro. Plants are a source of various biologically active molecules, e.g., strong antifungal activity has been shown in essential oil components [111]. Antifungal, antiviral, and antibacterial effects have been proven in many plant essential oils and their content substances, making them targets for active study [7,40]. Essential oils are hydrophobic mixtures of substances that are soluble in alcohol and ether. The chemical diversity of these ingredients ensures different mechanisms of action on pathogens. Out of several tested essential oils, the following plant genera appear to be significantly important: thyme (*Thymus*), wormwood (*Artemisia*), cinnamon (*Cinnamonum*), tea tree (*Melaleuca*), basil (*Ocimum*), resemary (*Rosmarinus*), fennel (*Foeniculum*), oregano (*Origanum*), clove (*Syzygium*), ajwain (*Trachyspermum*), myrtle (*Myrtus*), and others [112,113,114,115,116,117,118]. Several studies have documented the beneficial effect of essential oils in fighting *Candida* infections. The anti-biofilm activity of terpenes and the extraordinary effectiveness of carvacrol, geraniol, and thymol in the treatment of candidiasis associated with medical devices have been confirmed [119,120]. Excellent activity against the yeast *C. albicans* is shown by terpenoids, which can also be used in synergy with an antifungal agent, e.g., fluconazole [121]. Other compounds with antifungal activity obtained from plants are saponins, alkaloids, peptides, and proteins [115,117]. Due to its multi-resistance, *C. auris* is the most difficult to treat from the group of medically important *Candida* yeasts. The natural peptide crotamine, a toxin from rattlesnake venom, was used to successfully eliminate the yeast cells in vitro [122]. Other active agents against *C. auris* include phenolic compounds, nitric oxide nanoparticles, miltefosine and iodoquinol, and plant essential oils, of which carvacrol and farnesol are the most effective [123,124,125,126].

However, this approach is still an alternative treatment, which does not meet the conditions for use in the therapy of systemic yeast diseases.

## 6. Conclusions

Infections caused by yeasts of the genus *Candida* represent a veritable therapeutic problem, especially for immunocompromised patients. Currently, candidemia and related diseases are among the most serious fungal infections, with high morbidity and mortality. A serious factor in the virulence of some *Candida* species is the ability to form a biofilm, which ensures significant resistance to antifungal treatment by limiting the penetration of substances through the matrix, protects cells from the host’s immune responses, and represents a threat from the perspective of a permanent source of infection. Medically important species include *C. albicans*, *C. glabrata*, *C. tropicalis*, *C. parapsilosis*, *C. krusei*, and *C. auris*. They represent a risk, especially in the hospital environment, where they appear as nosocomial pathogens in chronically ill patients and postoperative patients. Due to their ability to adhere to inanimate surfaces of medical devices inserted into the patient’s body, *Candida* yeasts are a serious problem in medicine. Elucidation of the key mechanisms of biofilm formation is essential to advance the understanding and treatment of invasive *Candida* infections.

## Figures and Tables

**Figure 1 jof-09-00955-f001:**
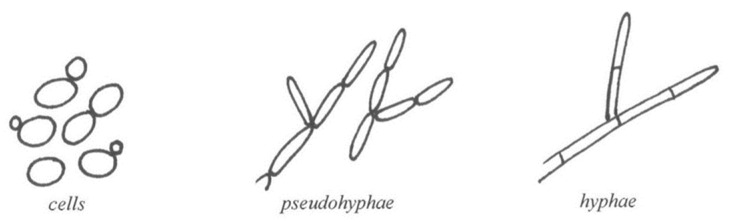
*Candida* cells, pseudohyphae, and true hyphae.

**Figure 2 jof-09-00955-f002:**
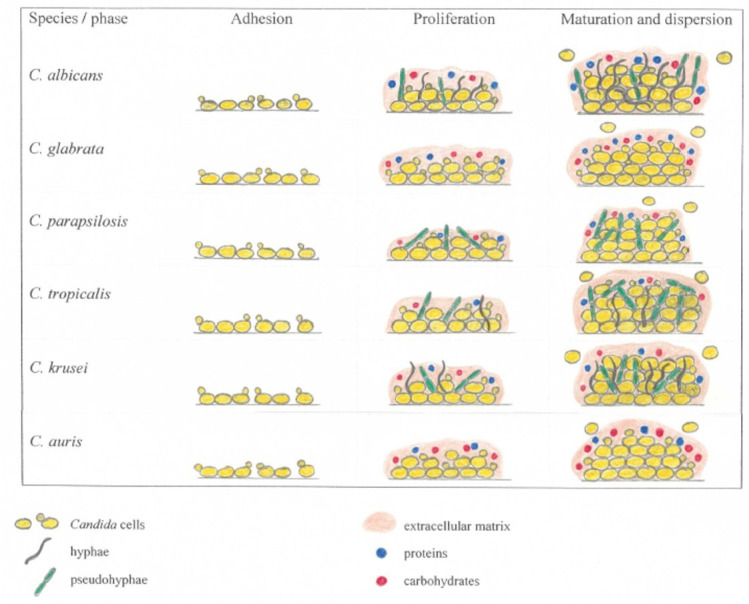
Biofilm formation in medically important *Candida* species. The illustration is only a schematic; it does not show actual cell sizes and spatial distribution of different morphological forms.

**Table 1 jof-09-00955-t001:** The most important recognized genes involved in the individual phases of *Candida* biofilm formation [86].

*Candida* spp.	Adhesion	Proliferation	Maturation	Dispersion
*C. albicans*	*ALS1*, *ALS2*, *ALS3*, *ALS4*, *ALS5*, *ALS6*, *ALS7*, *ALS8*, *EAP1*, *PGA10*, *RBT5*, *CSA1*	*CPH2*, *EFG1*, *TEC1*, *BCR1*, *HWP1*	*CPH1*, *TEC1*, *RBT1*, *NDT80*, *ROB1*, *BRG1*, *CZF1*, *GXF3*, *UME6*, *CPH2*, *ACE2*, *ZAP1*, *ADH5*, *GCA1*, *GCA2*, *CSH1*, *IFD6*, *FKS1*, *GAS1*, *GAS2*, *GAS3*, *BGL2*, *PHR1*, *XOG1*, *RLM1*	*PES1*, *UME6*, *HSP90*, *NRG1*
*C. tropicalis*	*ALS1*, *ALS2*, *ALS3*, *ALS4*, *ALS5*, *ALS6*, *ALS7*, *ALS8*, *ALS9*, *ALS10*, *ALS11*, *ALS12*, *ALS13*, *ALS14*, *ALS15*, *ALS16*	*ALS1*, *ALS3*, *EFG1*	*EFG1*, *RBT1*, *CZF1*, *GXF3*, *UME6*, *CPH2*, *FKS1*, *GAS1*, *GAS2*, *GAS3*	
*C. glabrata*		*EPA1*, *EPA6*, *EPA7*, *AWP1*, *AWP2*, *AWP3*, *AWP4*, *AWP5*, *AWP6*, *AWP7*	*FKS1*	
*C. parapsilosis*	*ALS1*, *ALS2*, *ALS3*, *ALS4*, *ALS5*, *PGA10*, *RBT5*, *CSA1*	*TEC1*, *BCR1*, *HWP1*	*TEC1*, *RBT1*, *NDT10*, *CZF1*, *GXF3*, *UME6*, *CPH2*, *ACE2*, *FKS1*, *GAS1*, *GAS2*, *GAS3*	

## Data Availability

Not applicable.

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
