# Peer review of "Biofilm Formation in Medically Important Candida Species"

_jof, 2023, doi:10.3390/jof9100955_

Round 1
Reviewer 1 Report
I found the present review well structured and usefull to have a quick overview on the topic of Candida biofilm.
I don't have major concerns. I only would suggest to give a focus on the diseases in which Candida species are involved (instead of taking in general of candidiasis). Specifically, a short section briefly describing the main pathologies (like VVC, oral candidiasis, biofilms on medical devices...) in which candida biofilms are relevant can be usefull.
English is fine (just check for typos).
Author Response
Dear reviewer,
The main goal of the submitted article is to provide an overview of biofilm formation by various Candida species, especially in relation to serious patient illnesses and contamination of medical devices introduced into the body, as stated in the article. Some infections of the organ systems are also briefly mentioned there. We have added the two most common candidiasis to them (oral candidosis and vulvovaginal candidosis), but we do not think that the infections you mentioned need to be described in detail.
Thank you.
Reviewer 2 Report
The manuscript of the review entitled “Biofilm formation in medically important Candida species” concerns the current problem of biofilm production by fungal pathogens. The article proposal presented by the authors has the potential for publication but requires major corrections.
The work contains too general information, largely without going into details. Individual issues should be broadened with their in-depth analysis, so that the main aim of this review is emphasized, because in the current form of the manuscript it is not clear what the authors wanted to highlight in presented comparisons.
The last section on biofilm resistance and treatment of biofilm-dependent infections is particularly interesting - this section should be expanded, it would be best to systematize information in the form of tables, paying particular attention to comparisons between species regarding the diversity of resistance mechanisms and susceptibility to alternative treatments.
Authors should consider the latest recommendations for the fungal nomenclature and include them in the manuscript. According to work by Borman, Andrew M, and Elizabeth M Johnson. “Name Changes for Fungi of Medical Importance, 2018 to 2019.” Journal of clinical microbiology vol. 59,2 e01811-20. 21 Jan. 2021, doi:10.1128/JCM.01811-20 Candida glabrata should be referred to as Nakaseomyces glabrata, and Candida krusei as Pichia kudriavzevii. It would be interesting to discuss interspecific differences in this context of distinct taxonomy.
Candida auris should also be included in the information set, as it is currently the Candida species of greatest global concern.
Figure 1 shows basic, well-known information, it does not contribute anything significant to the work.
Pichia kudriavzevii/C. krusei is usually found in two basic morphological forms, such as yeast and pseudohyphae, but not true hyphae, so the depiction in Figure 2 is misleading. In general, Figure 2 does not very vividly show the differences in biofilms between species, although in the text the authors write about different cell sizes, in the figure they are all the same, of the same shape, and additionally the hyphae or pseudohyphae are of a different color than cells and they are too small when compared to blastospores. The spatial distribution of different morphological forms in biofilms may also be different for Candida species, and this is not reflected in the scheme.
Furthermore, additional figures are needed to illustrate the presented issues, e.g., showing the involvement of virulence factors or regulatory genes in the various stages of biofilm formation by different species.
Table 1 contains information taken from another review - what is its novelty aspect or advantage over the previous work?
Why is the EFG1 gene not listed in Table 1 for C. albicans?
Genus and species names should be italicized thorough the text.
Thorough language proofreading is necessary due to several grammatical and stylistic mistakes.
Author Response
Dear reviewer,
The main goal of the article is to provide an overview of biofilm formation by various Candida species, and its characteristics. The article is intended for a wide range of readers, especially clinicians and scientists from various fields of research, not only experts who are primarily interested in the biology and pathogenicity of Candida in a certain area (e.g. genes, mechanisms of resistance, etc.).
The article was supplemented with information about C. auris.
We consider Figure 1 important for explaining the differences in morphological forms, as only experts are able to recognize them.
Figure 2 shows a diagram of biofilm formation at individual stages, including whether hyphae and pseudohyphae are present. The image is illustrative, the picture is only schematic, and it does not show the actual cell sizes or the actual spatial arrangement (this information will be supplemented in the article).
We cited the article written by Manuela Gómez-Gaviria and Héctor M Mora-Montesto, who reports that "C. krusei shows thermodimorphism, producing hyphae when growing at 37°C and blastoconidia and pseudohyphae when incubated at lower temperatures."
Table 1 was modified according to the picture and contains an overview of individual genes involved in a specific phase of biofilm formation in Candida species.
Table 1 was modified according to the figure given in the cited article No 59 and contains an overview of individual genes involved in a specific phase of biofilm formation in Candida species.
The EFG1 gene was accidentally omitted for C. albicans.
Thank you.
Round 2
Reviewer 2 Report
The authors addressed most of the comments, but still I have a few more suggestions.
If the authors wish to maintain the character of an introduction to discussing biofilm for the manuscript, this should be included in the title by adding an appropriate term, for example: "An introductory overview to.." or the like.
It should be "glabrata" not "glabrataa" in line 46.
In line 49 it should be C. auris.
In a "non-albicans" name, the species name "albicans" should be in italics.
The gene abbreviations like ALS or EPA should be in italics (for example in section 2.1).
C. auris biofilm should also be introduced in Figure 2.
The statements in line 280-283 do not refer to C. auris but to C. albicans biofilm, if the former does not form biofilms so strongly, it is difficult to conclude about the function of the matrix.
The sentence on lines 285-286 should have been moved earlier in the manuscript when the authors begin characterizing C. auris.
Additionally, the authors need to check the manuscript language for correctness due to several errors (i.e. line 58 and others).
The authors need to verify the language correctness.
